# Quaternary Ammonium Salts-Based Materials: A Review on Environmental Toxicity, Anti-Fouling Mechanisms and Applications in Marine and Water Treatment Industries

**DOI:** 10.3390/biom14080957

**Published:** 2024-08-07

**Authors:** Paola Marzullo, Michelangelo Gruttadauria, Francesca D’Anna

**Affiliations:** 1Department of Biological, Chemical and Pharmaceutical Sciences and Technologies (STEBICEF), University of Palermo, Viale delle Scienze, 90128 Palermo, Italy; michelangelo.gruttadauria@unipa.it (M.G.); francesca.danna@unipa.it (F.D.); 2Sustainable Mobility Center (Centro Nazionale per la Mobilità Sostenibile—CNMS), Via Durando 39, 20158 Milano, Italy

**Keywords:** quaternary ammonium salts (QASs), ionic liquids (ILs), environmental toxicity, anti-biofouling materials, marine anti-fouling systems

## Abstract

The adherence of pathogenic microorganisms to surfaces and their association to form antibiotic-resistant biofilms threatens public health and affects several industrial sectors with significant economic losses. For this reason, the medical, pharmaceutical and materials science communities are exploring more effective anti-fouling approaches. This review focuses on the anti-fouling properties, structure–activity relationships and environmental toxicity of quaternary ammonium salts (QAS) and, as a subclass, ionic liquid compounds. Greener alternatives such as QAS-based antimicrobial polymers with biocide release, non-fouling (i.e., PEG, zwitterions), fouling release (i.e., poly(dimethylsiloxanes), fluorocarbon) and contact killing properties are highlighted. We also report on dual-functional polymers and stimuli-responsive materials. Given the economic and environmental impacts of biofilms in submerged surfaces, we emphasize the importance of less explored QAS-based anti-fouling approaches in the marine industry and in developing efficient membranes for water treatment systems.

## 1. Introduction

Biofouling is the undesirable attachment of microorganisms to inanimate material followed by the secretion of by-products including polysaccharides and metabolites to form an extracellular matrix (EPS). The multicellular community and its matrix constitute the biofilm [1].

The formation of bacterial biofilms leads to significant economic losses in the marine, construction, food, and agricultural industries [2,3]. Similarly, in a hospital setting, biofilms of opportunistic pathogens (*E. coli*, *P. aeruginosa*, *S. aureus* and *S. epidermidis*) and fungi on surfaces and medical devices increase the risk of serious nosocomial infections [4,5].

Historically, the first anti-fouling approach consisted of biocidal compounds that killed or inactivated bacteria in disinfectant and sterilant treatments. Among these, quaternary ammonium salts (QASs), largely discussed in this review, play a key role [6].

Today, the widespread phenomenon of antibiotic resistance and the development of highly resistant biofilms has posed a major challenge to researchers [1,7,8]. Particularly, the extensive use of QASs is related to the emergence of resistance mechanisms among pathogenic organisms [9]. Expressing efflux pumps and altering fatty acid and protein composition are the main resistance mechanisms found in bacterial cells [10,11,12,13].

As a result, new approaches to blocking the formation of biofilms appear to be the most promising in the materials science community.

Specifically, passive or active anti-fouling systems can be identified according to their mechanism of action (Figure 1). Passive approaches include (i) non-fouling materials that can minimize surface interactions with proteins, bacterial cells or other biomolecules via a hydration layer formation (e.g., polyethylene glycol or zwitterionic polymers); (ii) low surface energy anti-fouling materials that can reduce biofuel adhesion forces (i.e., silicone, fluorine and slippery liquid-infused porous surfaces (SLIPSs)); and (iii) leachable polymer materials that release biocidal compounds into the environment.

Active approaches include non-leaching polymer materials, known as contact-killing surfaces.

Recently, Yao et al. reviewed the synthetic methods that use QAS-based composites and classified them by chemical nature (organic or inorganic) [14].

In this study, after a first part focused on the structure–activity relationship and toxicity of QASs, we reviewed recent developments in QAS-based anti-fouling materials by classifying them according to their mechanism of action. Moreover, we report scientific evidence on new and green anti-fouling systems that exploit the biocidal activity of QASs, often combined with non-fouling and foul-release properties. We also highlight QAS-based materials with stimuli-responsive properties. Therefore, the term anti-fouling is used not only to indicate the antibacterial activity of the QASs, but rather to also describe complex systems capable of counteracting microbial attack or killing them on contact to suppress biofilm formation.

Several papers report biomedical applications of QAS-based antimicrobial polymers as valid alternatives to antibiotics, with long-term protection and no resistance phenomena [15]. On the other hand, the development of QAS-based materials for marine industry and water treatment remains underexplored. Therefore, the second part of this paper focuses on approaches that can significantly inhibit bacterial adhesion to submerged surfaces.

## 2. Antibacterial Activity of Quaternary Ammonium Salts (QASs) and Ionic Liquids (ILs): Structure–Activity Relationship and Toxicity Studies

Quaternary ammonium salts (QASs) have been studied as disinfectants since the 1930s, when Domagk demonstrated the antimicrobial properties of benzalkonium chloride [16]. Like other molecules, they act by destabilizing the structure of the bacterial cell membrane and are known as membrane-active biocides [17]. The common structural feature of all QASs is a positive nitrogen with a long alkyl chain. QASs are classified as mono-QASs, bis-QASs, also known as Gemini-QASs, and poly-QASs, depending on the number of charged nitrogen atoms (Figure 2a) [18].

QASs are now widely used in the antimicrobial market as cationic surfactants in disinfectants, pharmaceuticals, hygiene products and cosmetics (Figure 2b) [19,20].

In addition, biocidal QASs have a significant role in water treatment and the paint industry [21,22].

**Figure 2 biomolecules-14-00957-f002:**
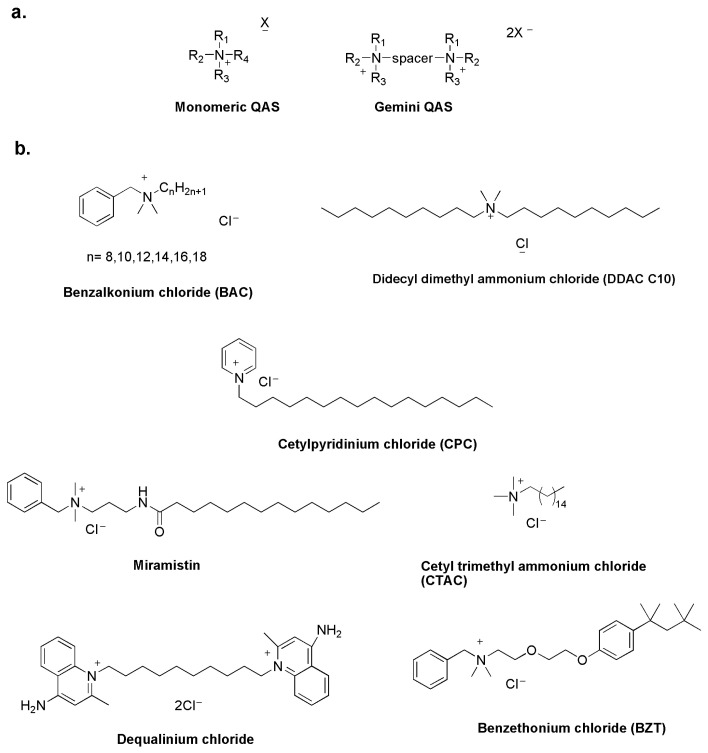
General structure of QASs (**a**) and commercial QAS-based disinfectants (**b**) [23].

QASs’ activity depends on the pathogen type and cell wall structure. There are significant differences between Gram-positive and Gram-negative bacteria, mycobacteria, viruses and fungi [24,25,26,27]. QASs interact with the negatively charged phospholipid via positive nitrogen, replacing the physiological stabilizing cations (Mg^2+^ and Ca^2+^). The hydrophobic tail then penetrates the hydrophobic core of the bacterial cytoplasmic membrane and the outer membrane of Gram-negative bacteria [28,29]. Exposure to QASs results in membrane disorganization with the release of cell components, such as lytic enzymes, autolysis and the loss of intracellular material [30,31]. However, the mechanism of action of QASs remains unclear. Some derivatives show binding to intracellular targets such as DNA and a ROS (reactive oxygen species) response [32,33]. Furthermore, a QAS’s biocidal activity is strongly related to the chemical structure of the cationic hydrophilic head and hydrophobic alkyl chain. Their antibacterial activity increases with the length of the alkyl chain, from three to sixteen atoms [34]. An N-alkyl chain with 12–14 carbons provides optimal antibacterial activity against Gram-positive bacteria, while a hydrophobic chain with 16 carbons is more effective against Gram-negative pathogens [35].

Recent works analyze the structural design of antimicrobial QASs and evaluate the biocidal activity of the ionic liquid (IL) class [36]. Note that the ionic liquid compounds considered in this review fall under the QAS classification with the characteristic of a low melting point (<100 °C). Many combinations of cations and anions allow us to focus on structure–activity relationships to identify new antimicrobial agents against multidrug-resistant strains [37,38,39]. Furthermore, ILs are attractive solvents or catalysts for several industrial applications [40]. Among the ILs, the classical ones, based on imidazolium cations, have been extensively studied for their antimicrobial and antibiofilm activity against many bacteria and fungi [41,42]. Unfortunately, the widespread use of water-soluble QASs and ILs can lead to their accumulation as aquatic pollutants. The result is a toxic effect on water organisms through indiscriminate interaction with all bio-membranes [23,43]. Thus, changing their chemical structure can increase hydrophobicity and reduce environmental toxicity [44]. As the cationic head and hydrophobic chain are critical in determining biocidal and cytotoxic activity, this remains a challenging area for research [45]. Focusing on recent advances in more selective antimicrobial surfactants, Zhou et al. analyze cationic surfactant self-assemblies or QASs inspired by immune system components (host defense peptides—HDPs) [36].

Generally, if their positively charged nitrogen is part of an aromatic ring (pyridine, imidazole), compounds are more toxic than those based on non-aromatic cations (piperidine, pyrrolidine and morpholine) [46].

IL toxicity studies on various cell cultures, vertebrates and invertebrates show a direct correlation with cation side chain length [47]. In this regard, several papers report the acute toxicity of imidazolium ionic liquids [48,49]. In inhibition tests against microorganisms, a low level of biodegradability is shown. Furthermore, the toxicity is not dependent on the type of anion but increases with the length of the n-alkyl chain [50]. Acute toxicity studies with marine and freshwater fish helped identify nontoxic and promising 1-butyl-3-methylimidazolium ILs for industrial processes [51].

On the other hand, the bioconcentration of imidazolium ILs in the tissues of marine invertebrates has been demonstrated for the first time in a recent study. The 1-methyl-3-dodecylimidazolium ILs salt shows increased bioaccumulation in *Mytilus trossulus* mussels related to the side chain length. Given the high concentration required for acute toxicity, the potential environmental risk may be due to high persistence and bioaccumulation [52].

In addition, derivatives with oxygenated functional groups in the side chain are less cytotoxic when assessing toxicity [47]. Hydrophilic anionic motifs (chlorine, bromide, formate, acetate and lactate) with lower toxicity are of interest in the search for safe disinfectants [53].

## 3. QAS-Based Anti-Fouling Materials

### 3.1. Non-Leaching and Contact-Killing QAS-Based Polymers

In the design of biocidal polymers, both quaternary ammonium and quaternary phosphonium salts are widely used. These materials have a high cationic density, which enhances the interaction with the negatively charged bacterial cells and causes them to die (Figure 1) [54,55]. In addition, programmed cell death signaling involving affected microorganisms on the surface and those in the vicinity can ensure long-lasting antibiofilm activity [56].

Surfaces of industrially used plastic polymers such as polyethylene or polypropylene have been effectively modified with quaternary ammonium salts to obtain anti-fouling materials [57,58]. However, the scientific community is now turning to QAS-based polymers derived from natural sources as the main alternative to those based on petrochemicals [59].

Quaternized chitosan polymers, shown in Figure 3a,b, have been extensively investigated for preparing biomedical materials such as self-healing hydrogels for tissue engineering and wound treatment [60,61,62,63,64].

Hydroxypropyltrimethylammonium chloride chitosan (HACC) (Figure 3c) is immobilized on the surface of various biomaterials such as titanium, hydroxyapatite or polymethylacrylate bone cement to prevent implant-related infections in both orthopedic and dental surgery [65,66,67,68,69].

Hydroxyl groups of sodium alginate, a cell wall component of marine brown algae, were coupled with organosilane quaternary ammonium salts to create a biocompatible, antibacterial polymer [70]. A recent study reports the covalent modification of a nanofibrillated bacterial cellulose (NFBC) with 2,3-epoxypropyltrimethylammonium chloride. NFBC shows excellent dispersibility, mouldability, biocompatibility and antibacterial activity. These properties make it suitable as a nanofiller in biomaterials or water-based coatings [71].

Poly(vinyl alcohol) (PVA) has been studied by several researchers due to its biocompatibility and biodegradability. The mixture of PVA and water-soluble lignin-quaternary ammonium salt was electrospun into nanofibers whose morphology resembles the extracellular matrix of the skin [58]. The quaternary ammonium compound epoxypropyl N,N-dimethyl dodecyl ammonium chloride has been grafted onto hydroxyl groups of polyvinyl alcohol formaldehyde (PVF) (Figure 4a) to produce an antibacterial macroporous sponge with water absorbency, good mechanical properties and biocompatibility [72].

QAS-based polyurethane polymers are one of the most studied antibacterial materials [73,74,75,76]. Xu et al. developed a synergistic antibacterial waterborne polyurethane (WRPUs) material with QASs and L-Arginine (Figure 4b). The charged guanidinium head group of arginine can form hydrogen bonds with negatively charged cell membranes and damage them [77]. Gemini-QASs with contact-killing ability were introduced onto the surface of a waterborne polyurethane polymer with self-polishing properties (Figure 4c). This coating can be biodegraded into non-toxic compounds, removing adhering microorganisms and renewing the surface [78].

**Figure 4 biomolecules-14-00957-f004:**
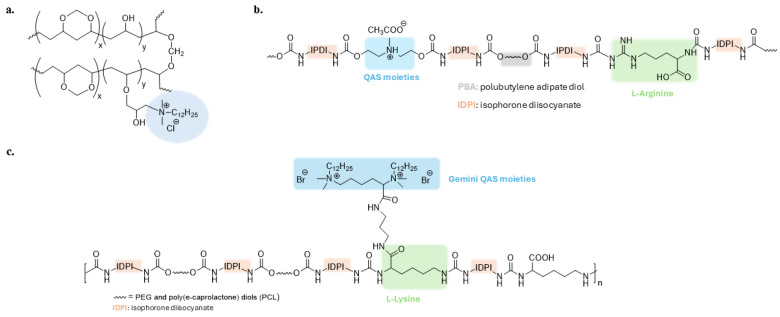
(**a**) Antibacterial polyvinyl alcohol formaldehyde (PVF) polymer. Adapted from [72]; (**b**,**c**) Polyurethane polymers grafted with quaternary ammonium groups. Adapted from [77,78].

Attractive organosilicon compounds include polysiloxanes and polysilsesquioxanes [79,80,81,82,83]. Polysiloxanes provide optimal surface properties and chain flexibility for close contact between linked functional groups and the bacterial cell membrane [84,85]. In the search for new antimicrobial polymers, polysiloxanes with pendant biocidal N,N′-dialkylimidazolium salt (ImS) (Figure 5a) prove to be potent antibacterial agents against several bacterial strains with high thermal stability in comparison to quaternary alkylammonium functionalized polymers [86].

Because a single molecule can be functionalized with multiple quaternary ammonium moieties, there is growing interest in polyhedral oligomeric silsesquioxanes (POSSs). The resulting materials have exhibited multiple cationic centers, good antimicrobial activity and compact size [87,88]. POSSs with a low degree of quaternarization and long alkyl chains have been investigated as antimicrobial additives for polysiloxane coatings (Figure 5b) [89]. Additionally, 1,2,3-triazolium was chemically attached to the surface of hydrophobic POSS nanoparticles to create a highly effective bactericidal additive for dental restoratives (Figure 5c) [90].

**Figure 5 biomolecules-14-00957-f005:**
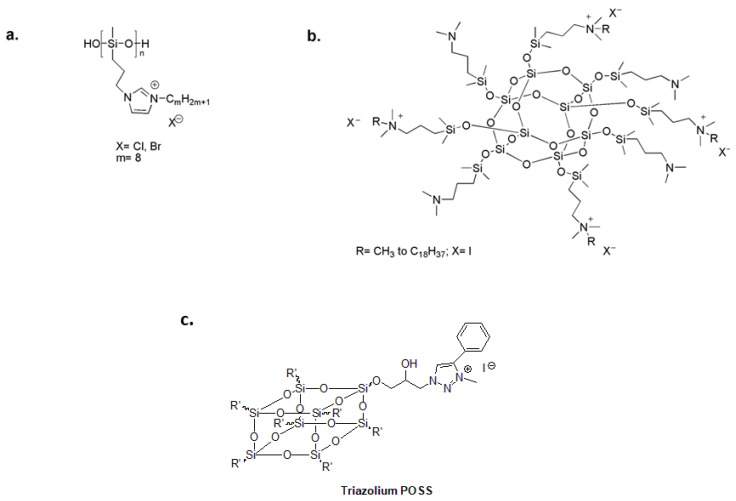
(**a**) Antibacterial polysiloxane polymers with imidazolium salts. Adapted from [86]. (**b**) Quaternary ammonium-functionalized POSS as an antimicrobial additive for polysiloxane coatings. Adapted from [89]. (**c**) 1,2,3-triazolium-functionalized POSS additive for dental restorative system. Adapted from [90].

### 3.2. Dual-Functional QAS-Based Anti-Fouling Materials

QAS can be covalently bound to the polymer chain or electrostatically bound to polyanions to provide non-leaching or leaching biocide polymers [91]. The first class of polymers, described in the previous paragraph, retain their bactericidal effect and do not release active substances into the environment [92]. With the second type of polymer, the rapid leaching of the biocide results in a high attack dose and reduced coating life. There is still little work on more complex materials that combine a variety of mechanisms of action. In a multi-defense strategy, a nitric oxide donor with diffuse antibacterial activity has been combined with a benzophenone-based quaternary ammonium antimicrobial (BPAM) immobilized on the surface of the Carbosil polymer (Figure 6a) [93].

Druvari et al. design new materials based on co-reactive polymers cross-linked by carboxylic acid or epoxy groups. In addition, the polymer substrates show both immobilized and released quaternary ammonium units responsible for better performance against *S. aureus* and *P. aeruginosa* (Figure 6b) [94].

Frequently, the antimicrobial activity of QASs is combined with the non-fouling or foul-release properties of superhydrophobic or hydrophilic materials [95,96,97,98,99,100,101]. Recent approaches combine a biocide sub-layer with a non-fouling upper layer [102]. However, He et al. developed an inverted surface in which a polyurethane polymer exhibits Gemini quaternary ammonium salts at the polymer/air interface. Underneath is a hydrophilic layer consisting of lysine and PEG residues. Isophorone diisocyanate (IPDI) and polytetramethylene glycol (PTMG) constitute the last hydrophobic layer (Figure 7a) [103].

Electrospun PVA nanofibers with a high specific surface area and an abundance of hydroxyl groups have been grafted with antibacterial QASs and highly biocompatible zwitterionic sulfopropyl betaine (ISB) with non-fouling properties (Figure 7b) [104]. Furthermore, Zhou et al. prepared a chitosan derivative with antibacterial activity guaranteed by isocyanate-terminated QASs and zwitterionic sulfopropylbetaine, compensating for the low biocompatibility of quaternized chitosan (Figure 7c) [63].

Cationic amphiphilic polymers containing hydrophobic alkyl chains and positive charges are extensively studied due to their similarity to antimicrobial peptides [105]. In these materials, the optimal balance between hydrophobicity and hydrophilicity influences antibacterial activity and biocompatibility [106]. Therefore, Zhao et al. investigated the antibacterial activity of amphiphilic copolymers containing quaternary ammonium groups and PEGylated alkyl chains of different lengths to improve hydrophilicity and biocompatibility (Figure 7d) [107].

A triblock copolymer consisting of PEG units, fluoro- and QAS-based polyacrylates has been developed to provide simultaneous non-fouling, foul-release and bactericidal activity (Figure 7e) [108].

**Figure 7 biomolecules-14-00957-f007:**
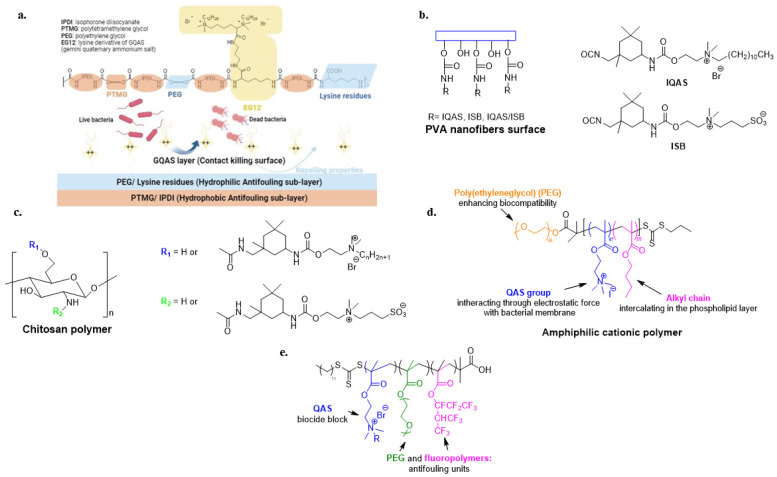
Representative examples of dual-functional polymeric materials. (**a**) Polyurethane with a contact-active antibacterial upper-layer and anti-fouling bacterial repellent sub-layer. Adapted from [103]. PVA (**b**) and chitosan polymers (**c**) modified with biocidal QASs and non-fouling/biocompatible zwitterionic group. Adapted from [63,104] (**d**) Amphiphilic cationic polymer with enhanced biocompatibility. Adapted from [107]. (**e**) Tri-block copolymer with anti-fouling and bactericide units. Adapted from [108].

The contact-killing antimicrobial performance of QASs has recently been coupled with the anti-adhesion ability of slippery liquid-infused porous surfaces (SLIPSs). Zhang et al. developed a polystyrene-based porous structure through a microphase separation technique using poly(ethylene glycol) (PEG) as a sacrificial template. The surface was then covalently modified with bactericidal 3-(trimethoxysilyl)propyl dimethylundecylammonium chloride and infused with silicon oil. This coating, which can be applied to a wide range of substrates, showed super-repellency against Gram-negative and Gram-positive bacteria. It retained its contact-killing ability even after the lubricating oil was depleted [109].

It should be noted that ionic liquids (ILs) have been shown to be promising lubricants in a variety of base oils [110]. Furthermore, the presence of structural motifs responsible for antibacterial and lubricant properties in the same molecule could suggest their use as additives in the design of slippery liquid-infused porous surfaces (SLIPSs).

### 3.3. Stimuli-Responsive and Switchable QAS-Based Materials

In recent years, responsive materials that are sensitive to physical stimuli from the environment (i.e., temperature, light or electricity) and to bacterial metabolites have been developed as antibacterial platforms [111].

Contact-active and light-responsive antimicrobial polymers are a valid approach to combat the contamination of touchable surfaces. An innovative self-polishing material was developed to combine the biocidal activity of quaternized chitosan with the antibacterial activity of TiO_2_ nanoparticles stimulated by visible light irradiation [112]. Other studies have suggested combining QAS with photosensitizers. The latter, when exposed to visible light, generate reactive oxygen species (ROS) that destroy biomolecules and bacterial membranes of pathogens. For this purpose, Santos et al. mixed a waterborne polyurethane varnish with photosensitizer curcumin and antimicrobial homopolymers consisting of poly((aminopropyl)- trimethylammonium chloride) (PAMPTMA) or amphiphilic copolymers with poly(butyl acrylate) (PBA) [113]. Similarly, another study covalently modified biopolymer cellulose with protoporphyrin IX, a photosensitizer, and QASs (Figure 8a). The cellulosic polymer increases the water solubility of the hydrophobic porphyrin, while the electrostatic repulsion between the QASs prevents its aggregation. The resulting polymer is water soluble and biocompatible. Moreover, it can be applied to various substrates via spray coating [114].

A recent study reports a copolymer based on thermosensitive units of poly (N-isopropylacrylamide) (PNIPAM) and quaternized poly[2-(dimethylamino)ethyl methacrylate] (PDMAEMA). The PNIPAM units, upon heating or cooling, are responsible for a reversible phase transition to form a globular or helical structure. The authors investigate the antimicrobial and antibiofilm activity of the copolymer at a lower critical solution temperature (LCST < 30 °C) and below (LCST > 30 °C) [97].

Often, QASs’ antibacterial activity is conjugated with photothermal properties of molecules or materials [116]. When exposed to infrared light, they absorb light energy and convert it to heat. The resulting hyperthermia acts synergistically with the QASs, enhancing the initial membrane damage.

Lu et al. constructed a bactericidal coating on the surface of SEBS, a styrene block copolymer. They exploited the photothermal property of polydopamine (PDA) and the bactericidal activity of a poly(lysine) derivative with QASs (Figure 8b) [117].

Near-IR carbon dots with excellent photothermal performance have been modified with QASs to obtain antibacterial materials. Moreover, their inherent luminescence suggests potential bioimaging applications [118].

Another stimulus for smart materials with antibacterial properties is the local acidification of infected wounds. A mussel-inspired pH-responsive hydrogel with adhesive and antibacterial properties was prepared using quaternized chitosan (HTCC) and oxidized dextran/dopamine adduct (OD-DA) (Figure 8c). Crosslinking interactions consisting of Schiff base and catechol-catechol adducts degrade in an acidic environment, releasing antibacterial silver nanoparticles (AgNPs) and pro-angiogenic drugs [115].

Switchable materials have optimal properties to ensure bactericidal or bacteria-repellent functions in response to multiple stimuli, such as light, heat or *pH* [119,120,121]. New hydrogels exploit this principle and represent a powerful tool for biomedical applications [122,123]. A recent example is a new photo-responsive hydrogel containing photolabile and biocidal quaternary ammonium groups. In detail, the cationic form of polymer based on 2-((4,5-dimethoxy-2-nitrobenzyl)oxy)-N-(2-(methacryloyloxy)ethyl)-N,N-dimethyl-2-oxoethan-1-aminium(CBNA) monomer (PolyCBNA) kills the bacteria, then its photolysis leads to the formation of a zwitterionic polymer (Poly(carboxybetaine methacrylate- PolyCBMA) capable of releasing the debris on the surface (Figure 9a) [119].

Wu et al. developed a multilayer coating consisting of crosslinked dopamine-anchored poly(acrylic acid) (PAA-dopa) and chitosan quaternary ammonium salt (Q-CS). The carboxylic groups of PAA are pH-responsive with a deprotonated form at basic pH or with a polycationic biocidal form in an acidic environment (Figure 9b). This adaptive and intelligent antibacterial system could have applications in the treatment of recurrent and chronic wounds characterized by lower *pH* due to the acidic substances produced by bacterial metabolism [120].

### 3.4. QAS-Based Nanocomposite Materials

Nanomaterials with at least one component at the nanometer scale (1–100 nm) have been considered as emerging potential alternatives in antibiofilms. Their nanoscale nature can promote intimate interactions with bacterial cells and damage them through a poorly understood mechanism [124].

In addition, in the design of non-fouling and super-hydrophobic coatings, it is well known that the micro/nanostructures increase the surface roughness and hydrophobicity with few attachment points for fouling organisms (lotus effect) [125].

Inorganic nanoparticles (e.g., metal oxide NPs) and organic polymeric components are the most common components of these antibacterial nanomaterials. Antibacterial QASs are optimal candidates for nanocomposite design. More recent studies will be reported here and in the following sections concerning marine applications.

An organic–inorganic nanocomposite applicable as a cotton fabric finishing agent has been synthesized by combining silicone quaternary ammonium salt and Ag/ZnO nanoparticles [126].

Biocompatible quaternary ammonium chitooligosaccharides (QACOSs) were electrostatically assembled on the surface of inorganic ZnO/palygorskite to prepare an organic/inorganic nanocomposite with antibacterial activity. The presence of the QACOSs modulates the surface charges, ensuring close contact with bacterial cells [127].

Graphene oxide has been used as a support for dodecyl dimethyl benzyl ammonium chloride and bromohexadecyl pyridine in the preparation of a new antibacterial nanocomposite. QASs assembled on the surface via π–π interactions ensure prolonged antibacterial performance and low cytotoxicity [128].

He et al. proposed the use of quaternary ammonium cross-linked micelles as a stabilizer and template for antibacterial silver nanoparticles. In an acidic biofilm microenvironment, the nanocomposite material releases AgNPs and QASs that synergistically exert antibiofilm activity [129].

Cyclodextrin modified with quaternary ammonium salts (β-CDQA) and hydrophobic adamantane-terminated polyethylene glycol has been used as a surfactant and cross-linking agent to prepare a nano-emulsion of tea tree essential oil (TTO). In this way, the antibacterial effects of TTO and β-CDQA were conjugated in the same material [130].

In the last few years, carbon nanodots (CDs) conjugated with bactericidal moieties have been investigated in antibacterial strategies. A binary system of photocatalytic molybdenum disulfide nanoflowers and QAS-modified copper-doped carbon dots was developed by Gao et al. This new nanocomposite exerts a triple bactericidal activity through QAS-mediated membrane damage, ROS production and a photothermal effect [131].

Fluorescing and antibacterial carbon dot nanomaterials have been prepared from quaternary ammonium salt of chitosan (QCS) by a simple preparation method. Authors explore the influence of different experimental parameters like solvents or excipients to obtain best antibacterial performance. Materials called QCS-EDA-CDs prepared with ethylene diamine (EDA) exhibit antibacterial activity against *Staphylococcus aureus* with single oxygen species production under daylight lamp irradiation. Meanwhile, their low hemolytic rate and optical properties suggest using them for imaging biofilms [132].

In addition, QAS-based nanostructured polymers targeting the bacterial genome and membrane have been proposed as candidates for the control of drug-resistant bacterial infections in mice [133].

Li et al. synthesized quaternized cellulose (CBHE-QAn) by esterification with 6-bromohexanoyl chloride followed by reaction with tertiary amine. Then, a simple nanoprecipitation method exploiting electrostatic interactions was used to obtain hybrid nanoparticles with citric acid. Hybrid NPs and composite films with PVA exhibit antibacterial properties against Gram-positive *S. aureus* and Gram-negative *E. coli.* In addition, the nanofiller improves the mechanical properties of the films [134].

## 4. Materials for Water Treatment

QASs impart antibacterial and anti-fouling properties to filtration membranes, which are highly susceptible to biofouling, in water treatment processes. Specifically, surface modification is a fouling resistance strategy for commercial membranes. Recently, to improve hydrophilicity, anti-fouling ability and filtration rate, a cellulose acetate membrane was covalently modified by etherification with epoxy propyl dimethyl dodecyl ammonium chloride (EPDMDAC) (Figure 10a) [135].

An anti-biofouling polyvinylidene fluoride (PVDF) membrane was fabricated by introducing a hydrophilic interlayer of silica nanoparticles. The silica nanoparticles served as an initiator to graft QASs via atom transfer radical polymerization (Figure 10b). The modified membrane could have applications in wastewater treatment. The introduction of silica nanoparticles improved hydrophilicity and permeability, while QASs provided durable antibacterial activity [136].

Graphene-based materials are an alternative material for the design of gravity filters for water decontamination [137]. The antibacterial property of graphene oxide hydrogels was improved by combining them with quaternary ammonium compounds called piQAs. These compounds have long alkyl chains and a large aromatic portion (Figure 10c). The strong π–π interactions between the aromatic rings and the graphene lead to the formation of highly porous, stable, non-leaching surfaces with a high water treatment rate and antibacterial activity against Gram-negative (*E. coli*) and Gram-positive (*S. aureus*) cells [138].

**Figure 10 biomolecules-14-00957-f010:**
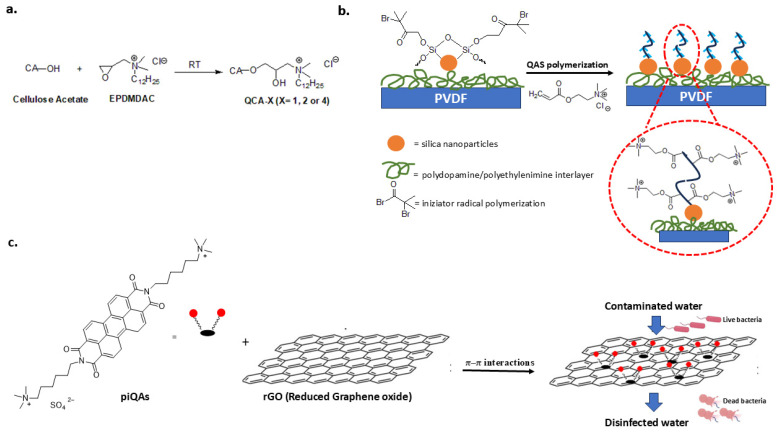
QAS-based materials for water treatment. (**a**) Cellulose acetate membrane covalently modified with QASs. Adapted from [135]. (**b**) PVDF membrane modified with QASs by silica nanoparticles interlayer. Adapted from [136]. (**c**) Assembly of aromatic quaternary ammonium compound (piQAs) with graphene hydrogel (rGO). Adapted from [138].

Chitosan quaternary ammonium salt (HTCC) has been used as a coagulating agent of algae-laden water and conjugated with a relatively moderate oxidant (sodium percarbonate SPC) in ultrafiltration pretreatment. In this approach, the electrostatic repulsion of algae cells, weakened by the cationic charge of HTCC, is combined with the change of electric potential on the surface caused by SPC [139].

An ultrafiltration polyethersulfone (PES) membrane with anti-biofouling properties was developed by Cihanoğlu et al. by depositing a low molecular weight surfactant (CTAB) and polydopamine (PDA) polymer. The PDA layer enhances the hydrophilicity of the PES membrane and provides biofouling resistance. CTAB, at a critical micellar concentration, imparts strong anti-biofouling properties. Depositing in the presence of nitrogen (N_2_) gas backflow mitigates the solution intrusion phenomenon while maintaining water permeability [140].

## 5. Marine-Biofouling and Anti-Fouling Approaches

Marine micro- and macroorganisms can colonize the surface of ship hulls or platforms, resulting in increased erosion, reduced ship maneuverability, increased fuel consumption, and environmental impact. Biofouling has a detrimental effect on marine transports and increases the cost of the maintenance required to ensure a ship’s service life [141]. Furthermore, fouled ships can spread marine organisms in new environments with biocontamination [142].

Five phases characterize the biofouling process (Figure 11). Within minutes, organic or inorganic molecules adsorb to the surface, forming a conditioning layer. Biofilm formation, creating an ideal substrate for the attachment of algae or protozoa, results from the subsequent colonization of unicellular bacteria and the secretion of an extracellular matrix (EPS). Finally, macrofouling organisms such as barnacles, mussels and bivalves grow on the surface [143].

Biofouling has been a major problem in several historical periods. The Phoenicians and the Carthaginians recognized the potential of copper for the prevention of biofouling in the hulls of ships. The Greeks and Romans introduced lead sheathing, which was adopted in the 18th century. With the introduction of iron ships, both of these approaches were abandoned due to their corrosive effects, and new non-metallic materials were investigated with no significant results. The search for biocide-releasing paint systems focused on copper, mercury or arsenic oxide dispersed in a soluble matrix that could be used as a primer [144,145].

In the 1950s, organotin compounds found a wide range of applications in the anti-fouling industry. However, in 2008, the International Maritime Organization imposed a worldwide ban on the use of tributyltin compounds due to their cytotoxic effects, bioaccumulation in water organisms and toxicity to the environment [146].

Despite their toxicity to non-target marine organisms and their persistence in the environment, anti-fouling paints based on copper compounds and booster biocides (e.g., Irgarol 1051, Diuron, Zinc Pyrithione) are the most used approach today [147]. Therefore, the production of biocide-releasing materials capable of releasing non-toxic compounds into the environment is a hot research topic. Based on the release mechanism, a distinction can be made between the insoluble or soluble matrixes. For permanent anti-fouling performance and lower toxicity, more environmentally friendly alternatives such as non-fouling and foul-release coatings are usually preferred [148]. Non-fouling materials are super-hydrophilic coatings, such as PEG or zwitterionic moieties, which can prevent the attachment of microorganisms by maintaining a hydration layer on the surface [149,150,151]. The foul-release systems with low surface energy favor the detachment of settled biofouling; as shown by the trend of the Baier curve, the retention of biological fouling is a function of surface energy [152]. Hydrophobic materials such as poly(dimethylsiloxanes), fluorocarbon or hydrocarbon polymers are in the range of minimum surface energy corresponding to a minimum of the curve. Therefore, these systems have low adhesive strength and a low modulus of elasticity. This favors the detachment of organic molecules or microorganisms due to the hydrodynamic flow caused by the movement of the ship [153].

### QAS-Based Materials for Marine Anti-Fouling Approaches

In marine anti-fouling research, as well as in the biomedical field, contact-killing surfaces with bacteria-releasing or non-fouling properties are alternatively investigated. In this sense, the use of quaternary ammonium compounds is still a little-explored area.

QASs that are used as paint additives are tested against marine bacterial microorganisms under the most natural conditions possible. In fact, the minimum inhibitory concentration (MIC) results for common laboratory strains are lower than those for marine communities composed of numerous Gram-negative pathogens. The length of the alkyl chains, the type of counterion and the presence of a ring in QASs are related to parameters measuring bacterial activity such as growth rate, total cell production and adhesion rate. A rapid decrease in adhesion rate is observed for compounds with less than four carbon atoms in the longest chain. Between 4 and 7 carbon atoms, cell production begins declining. As the length of the chain is increased by more than 7 atoms, a decrease in the growth rate has been observed. Iodide as a counter ion and phenolic ring on nitrogen are more effective [154].

An interesting study reports, for the first time, the anti-macrofouling activity of alkyl imidazolium ionic liquids [CnMIM]^+^[X]^−^ (Figure 12a) on dominant organisms such as barnacle larvae. Results indicate that [C12MIM][I] may be used in anti-fouling applications. Indeed, at nanomolar concentrations, it exhibits higher anti-biofouling and anti-settlement activity without lethal effects on non-target organisms. The activity of the tested compounds decreased with increasing alkyl side chains [155].

Moreover, hexyl imidazolium-based POSS (Figure 12b) adsorbed on the surface of biologically active oxides (Titania or 5 wt.% copper/titania) strongly prevents the adhesion of the marine environmental isolates of Pseudomonas PS27 and biofilm formation [87].

Polymer matrices or coatings that release quaternary ammonium compounds are promising marine biocides. A polymeric acrylate matrix (PMMA) has been incorporated with the antibacterial homopolymer poly(cetyltrimethylammonium styrene sulfonate) (PSSAmC16). An ion exchange process drives the release of the cetyl trimethyl ammonium cation (Figure 12c) [156].

Bellotti et al. formulated a new environmentally friendly paint consisting of rosin, a binder or plasticizer, and a biocidal quaternary ammonium “tannate” (Figure 12d). These can be obtained by the precipitation at *pH* = 4 or *pH* = 8 of tannins, natural polyphenols extracted from the native Argentinean tree “Quebracho”. The coating with rosin/oleic acid matrix and tannate derivatives shows an efficiency of ten months in sea water [157].

Marine anti-fouling literature reports QASs immobilized on crosslinked copolymers or grafted onto silica nanoparticles. To produce a hybrid anti-fouling/fouling release coating, polysiloxanes with pendant QASs are particularly attractive. In fact, the pendant biocide units lose their effectiveness due to the accumulation of dead microorganisms and macromolecules. Then, the non-fouling properties allow for a quick and easy cleaning of the surface [158]. In fact, the high flexibility of the polysiloxane chains can segregate quaternary ammonium groups to the surface/water interface. In addition, the anti-fouling properties prevent the formation of a layer of biomolecules or dead microbes, which deactivates the antibacterial activity of QASs. A synthetic process involves the binding by hydrosilylation of tertiary ammines (allyl dimethylamine) to a starting polymer consisting of polymethylhydrosiloxane (PHMS) and polydimethylsiloxane (PDMS), nitrogen quaternization with an alkyl halide and the use of vinyl PDMS as a crosslinker (Figure 13) [159]. Given the opposing properties of the hydrophobic polysiloxane matrix and the QASs, phase separation may be the cause of poor mechanical properties. Majumdar et al. applied a statistical experimental design to develop a QAS polysiloxane polymer with antibiofilm activity and adequate stability when immersed in seawater [160].

Several research groups have characterized the molecular structure of superficially quaternary ammonium-grouped PDMS using Sum Frequency Generation (SFG) spectroscopy [161]. The better anti-fouling activity of the QAS-tethered PDMS system from triethoxysilyl quaternary ammonium silane compared to those prepared from trimethoxysilyl quaternary ammonium silane has been justified by the large extension of the alkyl chain in the surface, as shown by SFG studies. The same result can be obtained with low molecular weight PDMS and QPOSS–PDMS systems with lower degrees of quaternization [162].

In QAS/PDMS systems, the water contact angle and surface roughness increase with the QASs’ concentration and molecular weight, producing heterogeneous surface morphology. The coating composed of QASs with a C18 alkyl chain and 18K-PDMS shows the best performance in fouling-release activity with higher surface heterogeneity and roughness [162].

Silicone rubber is one of the most widely used low surface energy materials for marine use. A recent paper reports on new films prepared by adding graphene and quaternary ammonium salts to silicone rubber. Water contact angle measurements show low surface energy for all systems. Anti-fouling performance, investigated in a marine environment, showed that only the graphene–silicone rubber composite had a durable anti-fouling effect due to deforming graphene and improved mechanical properties. The anti-fouling film with QASs has bactericidal action in laboratory tests but loses its function in the long term [163].

Recently, nanoparticles have been considered in anti-fouling coatings as functional nanofillers to provide antibacterial/anti-fouling activity with specific mechanical and optical properties [164,165].

In this field of research, silica nanoparticles (SiNPs) have attracted growing interest due to their high stability, good biocompatibility and easily modifiable functional groups [154,166,167]. Anti-fouling coatings with SiNPs are among the biomimetic approaches considered. They reproduce bioinspired surface topographies (e.g., shark skin) that are incompatible with microbial colonization [168,169].

It is also possible to conjugate silica nanoparticles with antimicrobial agents for targeted delivery [170,171]. Similarly, alkyl-functionalized silanes are widely used to introduce quaternary ammonium moieties that confer bactericidal and bacterial-repellent properties [172,173]. Indeed, long alkyl chains impart hydrophobicity to the coated surface with enhanced water contact angle and fouling-release activity.

SiO_2_ nanoparticles, functionalized with the bactericide dimethyl octadecyl [3-(trimethoxysilyl) propyl] ammonium chloride, were incorporated into a self-polishing acrylate polymer (Figure 14a) [174]. QAS groups on the surface of SiNPs increase compatibility for the acrylate polymer and no visible cracks are present.

In this system, SiNPs impart biocidal activity and micro/nanostructuring. The polishing rate is modulated by the content of triisopropyl silylmethacrylate (TISM).

Following immersion in seawater, hydrolysis of the TISM leads to increased surface roughness and super oleophobicity. As a result, the self-polishing surface is less susceptible to bacterial colonization. Moreover, the bactericidal activity increases with the TISM content due to the release of QAS-SiO_2_ into the bacterial suspension. In the natural field, after 14 days, only the high self-polishing coating removes the biofilm, maintaining the original micro-nano structure [174].

Mesoporous silica nanoparticles were also covalently modified with QASs, loaded with the biocide compound Parmetol S15 and added to paints to provide an antibacterial/anti-fouling effect (Figure 14b) [175,176]. Nanostructured coatings increased the hydrophobicity and surface roughness of PVC-coated panels. Passive protection was provided by QAS-modified silica and active protection by the controlled release of Parmetol S15. Compared to the pristine paint, dual functionalized nanoparticles provide antibacterial activity against *E. coli* and *S. aureus* and anti-macrofouling properties for mussel attachment without toxic effects on these organisms. After six months of exposure to the Red Sea, paints containing 2 and 5 wt.% of dual-functional nanoparticles reduced the accumulation of biofouling in comparison to the untreated system [175].

Marine biofouling is a major problem in aquaculture. Biofouling in nets reduces their life expectancy and results in high maintenance and production costs [177]. Lanioti et al. designed cross-linked antimicrobial polymers combining two copolymers poly(cetyltrimethylammonium 4-styrene sulfonate)-co-glycidyl methacrylate P(SSAmC16-co-GMAx) and poly(vinyl benzyl dimethylhexadecylammonium)-co-acrylic acid P(VBCHAM-co-AAx). The cross-linked polymeric network bear immobilized 4-vinyl benzyl dimethylhexadecylammonium chloride biocidal groups with releasable 4-styrene sulfonate cetyltrimethylammonium (Figure 15a). To create a strong coating for aquaculture nets, authors explored the hexamethylene diamine as a cross-linker and several blend combinations of copolymers [178].

Copolymers have also been used in a multilayer process using water as a green solvent [179]. The releasable biocide cetyltrimethylammonium is trapped in the inner layer with a slow and controlled release (Figure 15b). In the outer layer, the water-soluble poly(vinyl benzyl trimethylammonium chloride-co-acrylic acid) (PVBCTMAM-co-AA) is thermally cross-linked to the first layer via the carboxyl groups of the acrylic acid. For the outer layer, authors explore homopolymers of acrylic acid also combined with cationic poly(hexamethylene guanidine hydrochloride), chosen for its antimicrobial activity. In this way, the hydrophilic outer layer acts as a non-fouling surface and the immobilized 4-vinylbenzyldimethylhexadecylammonium imparts a contact-killing activity, while the leaching cationic groups (cetyltrimethylammonium and poly(hexamethylene guanidine hydrochloride) provide biocide-release properties. This antimicrobial coating shows high anti-fouling performance in real immersion conditions and easily removes fouling microorganisms with pressure washing [179].

Marine biofouling can reduce the performance and analytical accuracy of sensors and floats used to monitor water quality or aquatic species. Acrylate hydrogels loaded with dicocodimethyl ammonium chloride (Arquad 2C-75) have been effective in the prevention of micro- and macrofouling on the optical windows of marine sensors [180]. Recently, a polymeric antibacterial membrane sensor with anti-adhesive and regenerative properties has been developed. For this purpose, dual-functionalized microspheres of Fe_3_O_4_ were decorated with a non-adhesive zwitterionic polymer, poly(sulfobetaine methacrylate) (PSBMA), and grafted with QAS groups to confer antimicrobial properties (Figure 16). The anti-fouling magnetic Fe_3_O_4_ is magnetically repelled from the surface after fouling, ensuring the longevity of the marine sensor [181].

## 6. Conclusions and Future Perspectives

This review provides literature data on QASs and their application in antibacterial and antibiofilm systems.

We have analyzed more “green” approaches, such as polymers that release non-toxic and low-environmental impact antibacterial molecules with specific structural motifs. Indeed, structure–activity relationships for QASs and ionic liquid classes can help to design and synthesize safer and more environmentally friendly compounds.

The discussion has shifted to composites covalently modified with QASs due to the rapid degradation of biocide-releasing approaches and the need to develop more effective and longer-lasting coatings.

The anti-fouling approaches presented here, with applications in the marine and water depuration industries, highlight the multidisciplinary research in this field. The antibacterial and contact-killing activity of QASs is often conjugated with specific chemical and physical properties of surfaces to achieve foul-releasing or non-fouling properties.

Researchers from different disciplines developing new anti-fouling and environmentally friendly QAS-based systems could be inspired by the information gathered in this review.

## Figures and Tables

**Figure 1 biomolecules-14-00957-f001:**
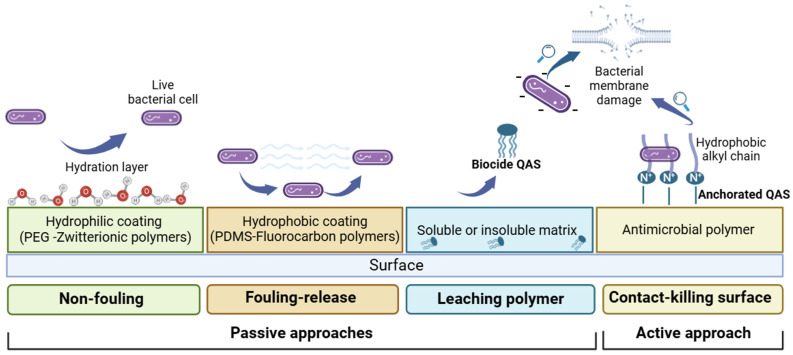
Schematic illustration of anti-biofouling strategies.

**Figure 3 biomolecules-14-00957-f003:**
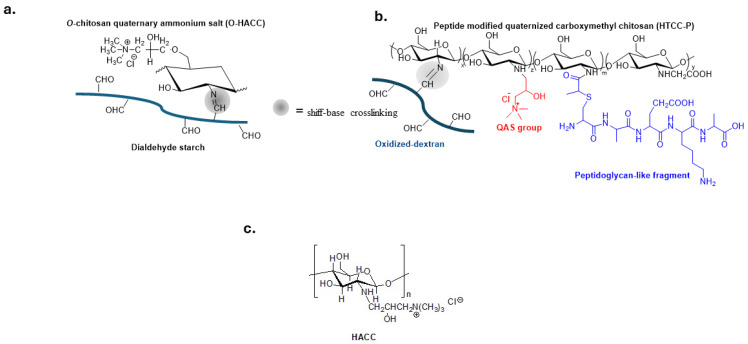
Bio-based polymers with QASs groups and reversible schiff-base crosslinking to guarantee self-healing properties (**a**,**b**). Adapted from [60,61]; (**c**) Hydroxypropyltrimethyl ammonium chloride chitosan (HACC) a biocompatible material in the orthopedic field.

**Figure 6 biomolecules-14-00957-f006:**
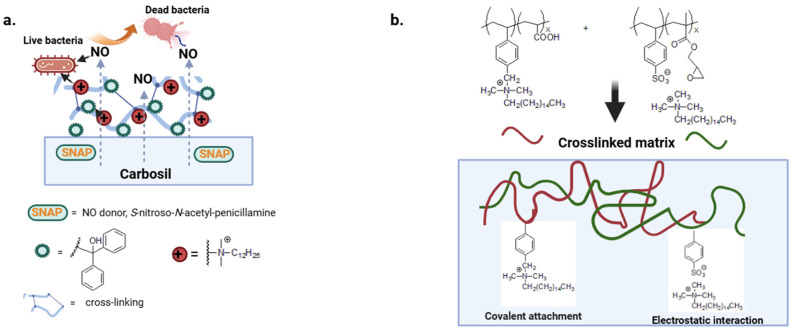
(**a**,**b**) Crosslinked matrix with diffusive and covalently attached biocide moieties. Adapted from [93,94].

**Figure 8 biomolecules-14-00957-f008:**
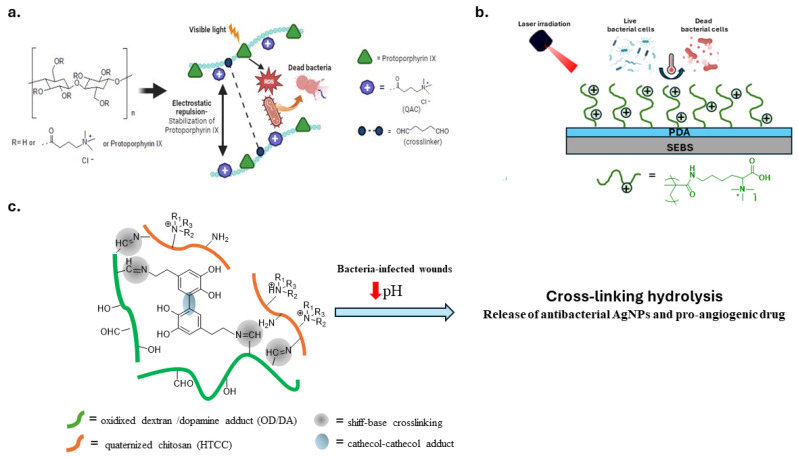
Representative examples of stimuli-responsive polymeric materials. (**a**) Cellulosic materials with antibacterial activity explicated by quaternary ammonium moiety and photosensitizers mediated ROS response. Adapted from [114]. (**b**) Polymeric platform with antibacterial activity explicated by lysine-QASs and polydopamine with photothermal properties. Adapted from [107]. (**c**) Cross-linked and pH-responsive hydrogel constituted of quaternized chitosan (HTCC) and oxidized dextran/dopamine adduct. Adapted from [115].

**Figure 9 biomolecules-14-00957-f009:**
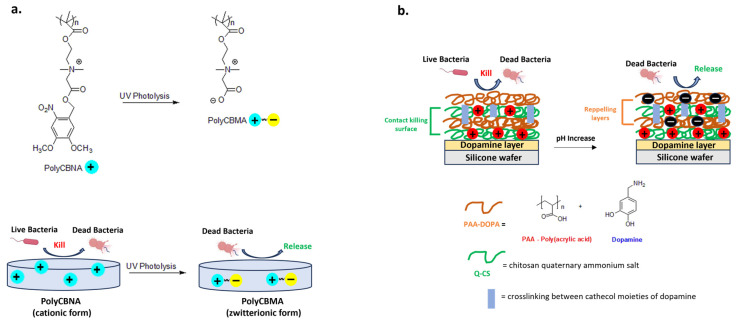
Representative examples of switchable materials. (**a**) Photo-responsive poly[2-((4,5-dimethoxy-2-nitrobenzyl)oxy)-N-(2-(methacryloyloxy)ethyl)-N,N-dimethyl-2-ox-oethan-1-aminium] (PolyCBNA) hydrogel switching from its cationic antimicrobial to zwitterionic anti-fouling form. Adapted from [119]. (**b**) Acid-sensitive multilayer with reversible charge’s surface and switchable bactericidal and bacteria-repelling functions. Adapted from [120].

**Figure 11 biomolecules-14-00957-f011:**
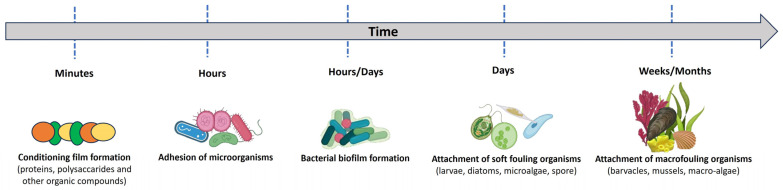
Schematic diagram of the marine biofouling process.

**Figure 12 biomolecules-14-00957-f012:**
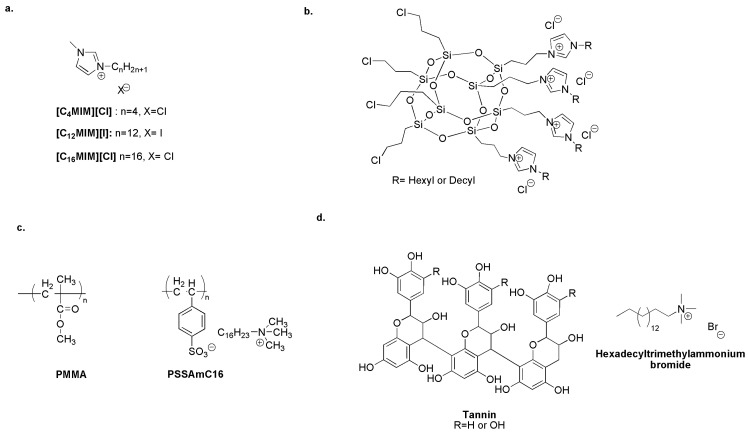
(**a**) Non-toxic anti-macrofouling alkyl imidazolium ILs. Adapted from [155]. (**b**) Imidazolium quaternized POSS adsorbed on the surface of biologically active oxides to generate an efficient anti-fouling system. Adapted from [87]. (**c**) Acrylate matrix incorporating the biocide releasing homopolymer PSSAmC16. Adapted from [156]. (**d**) Biocide quaternary ammonium “tannate” as an antimicrobial paint additive. Adapted from [157].

**Figure 13 biomolecules-14-00957-f013:**
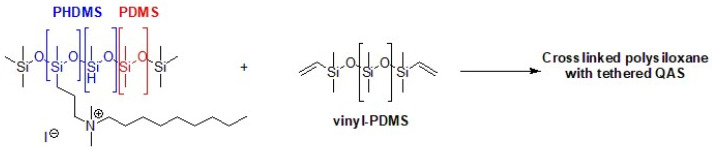
PHDMS–PDMS crosslinked polymer with pendant quaternary ammonium groups. Adapted from [159].

**Figure 14 biomolecules-14-00957-f014:**
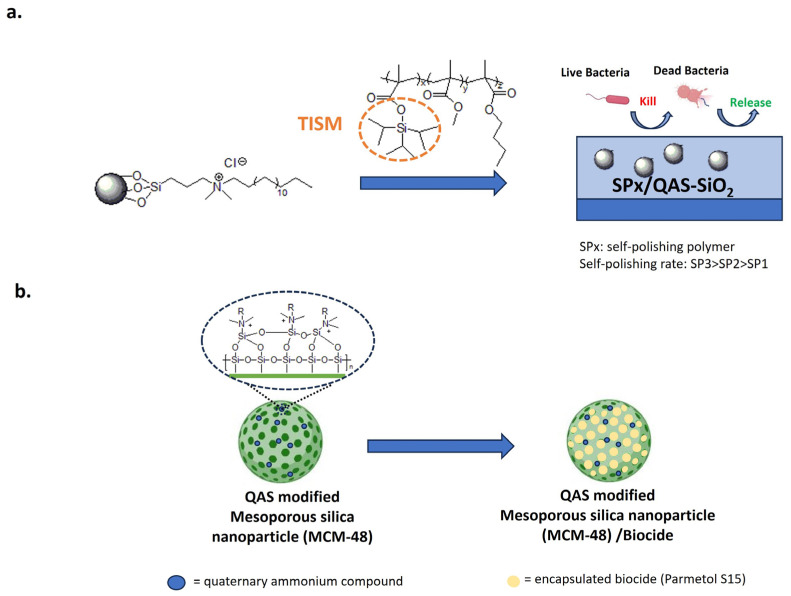
Anti-fouling approaches with silica nanoparticles. (**a**) Self-polishing acrylate polymer conjugated with antibacterial QAS-based silica nanoparticles. Adapted from [174]. (**b**) Mesoporous silica nanoparticles modified with contact-killing QASs and loaded with the biocide Parmetol S15. Adapted from [175,176].

**Figure 15 biomolecules-14-00957-f015:**
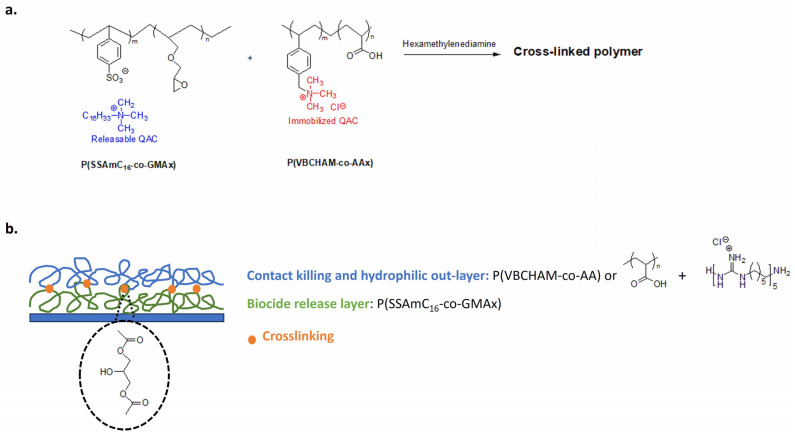
Antimicrobial materials for aquaculture nets. (**a**) Cross-linked antimicrobial polymer with releasable and contact-killing QASs. Adapted from [178]. (**b**) Multilayer coating with antibacterial and non-fouling polymers. Adapted from [179].

**Figure 16 biomolecules-14-00957-f016:**
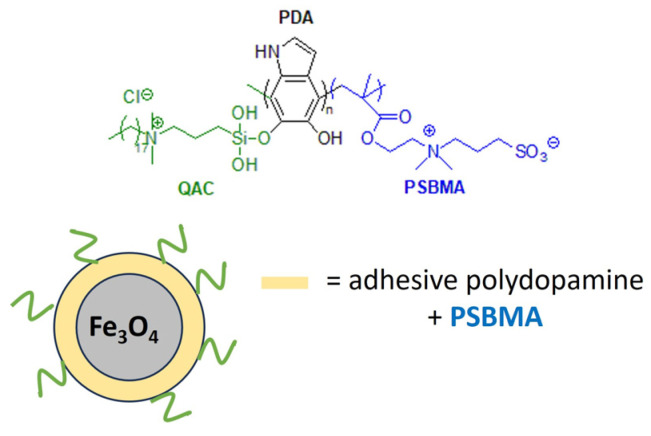
Anti-biofouling coating for marine sensors based on dual-functionalized magnetic composite. Adapted from [181].

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
