# Peer review of "Quaternary Ammonium Salts-Based Materials: A Review on Environmental Toxicity, Anti-Fouling Mechanisms and Applications in Marine and Water Treatment Industries"

_biomolecules, 2024, doi:10.3390/biom14080957_

Round 1

Reviewer 1 Report

Comments and Suggestions for Authors

The review paper ‘Quaternary ammonium salts-based materials: a review on environmental toxicity, anti-fouling mechanisms and applications in marine and water treatment industries’ submitted by Marzullo et al. fits the journal’s scope, offering a solid foundation on the topic of biofouling and antifouling strategies.

The main criticism is the overall flow of the review is challenging to follow. Currently, the aim and significance of the manuscript are discussed only after Section 2.

Here are some specific comments and suggestions for improvement

Ensure that each section transitions smoothly to the next. For instance, linking the introduction to the subheading

Line 38-39: should be one paragraph

Figure 2 appears fuzzy and some of the text is cut off

Line 148-149: following four sections of this paper? The review paper should be restructured to include the following 4 sections

Line 207: an adequate reference is missing

Line 246-250: it should be at least partly supported by reference

Line 302-303: Explain why this modification of protoporphyrin IX is significant

Line 303-307: enhance flow by combining the sentences

Line 347: acronym is not well-defined

The list of references should be prepared more carefully.

Author Response

Comment 1: The main criticism is the overall flow of the review is challenging to follow. Currently, the aim and significance of the manuscript are discussed only after Section 2.

Comment 2: Ensure that each section transitions smoothly to the next. For instance, linking the introduction to the subheading

Response 1 and 2: Thank you for pointing this out. To improve the flow of the review we have moved the content of section 2 of original manuscript and the Figure 2 (Pag. 3 - line 110-147) to the introduction of the revised manuscript (Pag. 1 - line 40-74). In addition, the sentence in line 38-39 has been added. This reorganization makes the purpose of the work immediately clear to the reader. A brief content description is also provided for subsequent sections.

Comment 3: Line 38-39: should be one paragraph.

Response  3: Thank you for your suggestion. Content from line 39 to line 54 (including Figure 1) of the original manuscript moved to start of next section (Revised manuscript, pag.2 -  line 77-90). Thus, all information related to structure-activity relationship and toxicity of QAS are in one paragraph.

Comment 4: Figure 2 appears fuzzy and some of the text is cut off.

Response 4: The figure has been corrected and replaced in the text as Figure 1 of the revised manuscript (pag. 2 – line 57).

Comment 5: Line 148-149: following four sections of this paper? The review paper should be restructured to include the following 4 sections

Response 5: In view of the restructuring of the introduction, which covers antifouling approaches in general (Revised manuscript - pag. 1 - line 40-74), sectiom 3 specifically review the different types of QAS-based materials (Revised manuscript - pag. 4 – From line 142)

Comment 6: Line 207: an adequate reference is missing.

Response 6: According with your suggestion references [84] and [85] have been added (Revised manuscript - pag. 6 – line 212)

Comment 7: Line 246-250: it should be at least partly supported by reference.

Response 7: According with your suggestion references [96-101] and [102] have been added (Revised manuscript - pag. 8 – line 250-251)

Comment 8: Line 302-303: Explain why this modification of protoporphyrin IX is significant.

Response 8: We have reported on the role of photosensitising agents such as protoporphyrin IX in analyzed systems. We detail the mechanism of action, which consists in producing ROS when exposed to visible light (Revised manuscript, pag. 9 – line 299-302).

Comment 9: Line 303-307: enhance flow by combining the sentences.

Response 9: Thank you for your suggestion. This part has been reviewed. (Revised manuscript,  pag. 10 – line 302-306).

Comment 10: Line 347: acronym is not well-defined

Response 10: Thank you for your suggestion. The names of the cationic and zwitterionic polymers with their respective acronyms have been inserted in the text as they appear in Figure 9a (Revised manuscript, pag 11 – line 346-350).

Comment 11: The list of references should be prepared more carefully.

Response 11: Thank you for your suggestion. This part has been reviewed.

Reviewer 2 Report

Comments and Suggestions for Authors

Quaternary ammonium salts and their derivatives are currently the most widely used biocides. The compounds are in aqueous and non-aqueous  solutions, in hybrid materials, as part of water-soluble polymers and synthetic polymers. The use  of these compounds in marine and water treatments industry is particularly important.

The title review gives a broad perspective on biocidal activity of alkylammonium derivatives and their application in industry. In addition to application prperties, the Authors pay a lot of attention to their biodegradability and impact on the environment.

The manuscript is well organized and well written based on well-chosen literature

Author Response

We would like to thank the reviewer for the kind report.